# Bridging Gaps, Fostering Inclusion: A Gendered Look at Disability Support for Women in Higher Education

Fatima Leon-Larios [1] , María-Luisa Benítez-Lugo [1,*] , Cristina Almendros-Caballero [1], Linnéa Stamatía Meyer [1], Isabel Corrales-Gutierrez [2,3,*] and Rosa Casado-Mejía [1]

1    Faculty of Nursing, Physiotherapy and Podiatry, University of Seville, 41009 Seville, Spain; fatimaleon@us.es (F.L.-L.); linmeyer19@gmail.com (L.S.M.); rcasado@us.es (R.C.-M.)
2    Surgery Department, Faculty of Medicine, University of Seville, 41009 Seville, Spain
3    Foetal Medicine Unit, Virgen Macarena University Hospital, 41009 Seville, Spain
*    Correspondence: marisabeni@us.es (M.-L.B.-L.); icorrales@us.es (I.C.-G.)

**Abstract:** This study investigates the effectiveness of inclusion measures for women with disabilities at a public university in southern Spain, aiming to understand their needs and experiences. Utilizing a gender perspective, the research engaged 12 women from various university roles in semi-structured interviews, later analyzed using NVivo 20, and organized into categories assessing institutional resources, working/academic conditions, and the interplay of disability and gender. Findings indicate that, despite no direct discrimination based on sex, the patriarchal social framework still fosters gender and inclusion gaps. Peer support emerges as a protective factor, whereas obstacles such as resource scarcity, access challenges, and a lack of awareness about disability measures pose risks. The study highlights the need for enhanced visibility of inclusive measures and the development of agile, individualized policies. It underscores the importance of raising awareness, particularly about non-visible disabilities, through educational initiatives.

**Keywords:** women; gender; inclusion; disability; high education; qualitative

## 1. Introduction

Researching inequality implies conducting an intersectional analysis, not only through stratification based on gender, race and ethnicity, social status, or sexual orientation, but also understanding that the origins of inequality are diverse and intertwined. This perspective aligns with the findings of Wolbring and Lillywhite (2021), who emphasize the need for comprehensive and critical engagement of disabled students, non-academic staff, and academic staff within equity, diversity, and inclusion (EDI) initiatives in universities. Their study revealed that effective inclusion must address multiple dimensions of identity and experiences of inequality to be truly impactful. Understanding women as a diverse group per se and focusing on functional diversity within higher education is essential to developing inclusive policies that recognize and address the complexities of inequality. Thus, it underscores the importance of enhancing visibility and developing agile, individualized policies to foster a truly inclusive educational environment [1].

According to the World Health Organization, around 15% of the global population has some kind of disability, specifically more than one thousand million people, according to UN Women [2]. However, despite all the advances in equality and equity, there are still great barriers to the social inclusion of people with disabilities [3], and the average prevalence rate in the female population aged 18 or over is 19.2% compared with 12% for men, which represents one in five women [2]. Moreover, women with disabilities face a double discrimination and inequality burden for being women and having a disability [3].

In higher education and the university spheres, there have been changes in the last few years regarding awareness, legislation, and the need to guarantee inclusion and access, resulting in an increase in students, administrative staff, and lecturers. Notwithstanding,

the report by the Spanish Committee of Representatives of Persons with Disabilities [4] indicates that the representation of people with disabilities in Spain is a concern, being just 1.5% of the total student population. Moreover, this percentage decreases in subsequent university years, reaching 1.2% in postgraduate and master courses and just 0.8% in Ph.D. studies. This student population shows a high rate of dropout in comparison to the general population [4]. This underscores the need for universities to not only focus on admission rates but also on retention and success strategies for students, administrative staff, and lecturers with disabilities.

Thus, this study combines gender, functional diversity, and educational level, although it must be considered that currently there is not a deep knowledge of the reality of women with functional diversity in the university. It was therefore necessary to understand their situations, needs, and experiences in relation to the services available to improve the social response in general, that of the University of Seville, Spain, and thus advance the agenda of real equality. The objective of this research was to explore the situation of women with disabilities who are part of the university community at the University of Seville, analyzing their risk factors and protection measures, as well as their difficulties and felt needs, as a way to offer a guide for institutional actions to be developed.

## 2. Materials and Methods

### 2.1. Study Design

After a bibliographical search in databases related to this subject, it is decided that a qualitative methodology will be applied to analyze in depth the experiences and opinions of the protagonists through their discourses in order to understand their life experiences, expectations, perceptions, and feelings. A descriptive, qualitative study is designed with a gender approach, establishing as units of observation women with disabilities (students, lecturers, and administrative staff) of the University of Seville community, and gender and disability, risk and protection factors, and requirements and difficulties as units of analysis. The study was conducted between January and July 2022. The COREQ criteria list was adapted to report the study's findings.

### 2.2. Participants and Sample Selection

A total of 12 participants were enrolled in this study, i.e., 4 students, 5 administrative staff, and 3 lecturers. To select the sample, key informants were contacted, i.e., individuals authorized to identify students and workers with disabilities who were contacted via email asking them to be part of this project. A convenience sampling was used to select participants considering the following inclusion criteria: women of legal age who were part of the US community, had any type of disability, were included in the university program for people with disabilities, and were able to communicate and understand the study requirements, therefore accepting and signing the informed consent.

After establishing a basic segmentation criterion for the collective they belong to within the university community, 3 types of profiles resulted from the following division: students, lecturers, and administrative staff. A minimum of 3 interviews per profile were planned, though it could be expanded if the degree of saturation was not reached. The following features or variables must be distributed: age, years at the institution, type of disability, area of knowledge, family/personal/close background regarding disability (indicating the type), and membership in any association for people with disabilities.

### 2.3. Techniques, Instruments, and Procedures for Information Gathering

The study conducted 12 semi-structured interviews with University of Seville lecturers, administrative staff, and students. None of the women contacted to join the project refused to participate. The technique selected was semi-structured interviews, as it fosters free expression on the part of participants and allows greater flexibility from the initial script to explore relevant topics when they arise.

The interview script designed contained open questions related to the study objectives, the bibliographic review conducted, and the opinions of key informants agreed upon by the research team. In total, they included 22–24 items, depending on the type of participant (female students/lecturers/administrative staff women).

The approximate duration of each interview was 40–50 min. After a personal introduction to the research, objectives, use of the information, and confidentiality, each participant signed her own informed consent. All interviews were recorded for their subsequent literal transcription and analysis. Due to the COVID-19 pandemic, interviews were conducted face-to-face, sometimes at the request of the interviewee, and online. They were carried out by two collaborators, nursing students at the University of Seville, trained in the execution of interviews and having no work or personal relationship with the interviewee. The interviews took place, when possible, at a location of the participant's choice. In situations where it was not possible to do so, the interview was conducted online, adapted to the circumstances and requirements of the interviewee. In these cases, the chosen format was the best-adapted videocall possible.

*2.4. Ethical Consideration*

This research complies with all the ethical guidelines and EU national and international legislation applicable regarding the protection of personal data, as well as relevant legislation regarding research.

Data processing follows a process of anonymity in which interviewees cannot be identified; each participant was assigned a fictitious name to preserve anonymity and guarantee data protection. The information and consent document signed by the participants contains all the information regarding data processing, and consent was explicitly asked in order to conduct data processing within the research to ensure transparency.

The data were stored in the cloud with an access code only known to the research team. Once the research and analysis end, these data will be deleted. This research has passed approval by the ethics committee at the University of Seville (1015-N-22).

*2.5. Data Analysis and Interpretation of Results*

Once the interview stage was completed, the information obtained was analyzed, assigning text fragments to the previously set categories and emerging categories. The information obtained was analyzed, assigning (Figure 1) text fragments to the previously set categories (institutional resources, working and/or academic conditions, disability, and gender) and emerging categories (subcategory gender role, reconciliation, and co-responsibility).

Transcribed interviews were analyzed through an inductive method of content analysis using the QSR NUD*IST Vivo20 using a thematic analysis with a gender approach. To this end, a systematic, iterative process was conducted following the phases described by the following [5]: (1) becoming familiar with the data through readings and notes; (2) codification; (3) elaboration of a thematic map; (4) identification of themes or categories; and (5) preparation of the report with the analysis of the selected data. This codified information was analyzed and compared with the scientific evidence currently available, with the aim of establishing conclusions and proposals that would improve the inclusion of people with disabilities at the University of Seville.

The triangulation and result debate with the team managing the care of people with disabilities at the University of Seville granted the study quality and validity. The structural analysis of the different parts of the study, researchers' reflexivity, and semantic and pragmatic triangulation (providing different perspectives from their fields of specialization, i.e., nursery, physiotherapy, anthropology, and gender studies) of data sources enabled validity. The study's transferability, confirmability, dependability, and credibility ensured its trustworthiness.

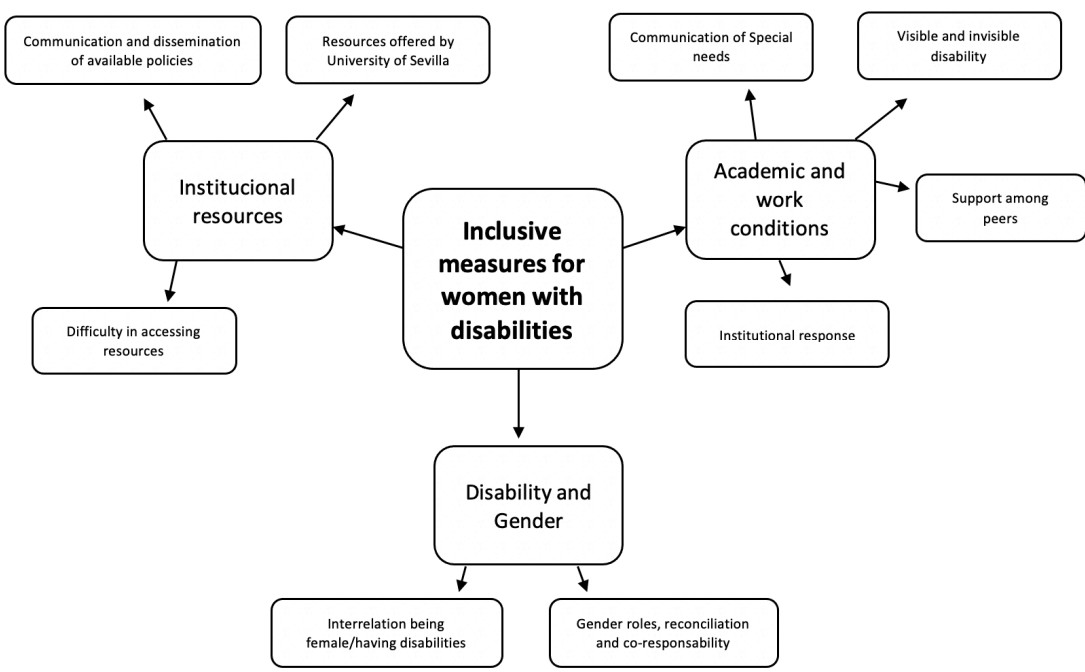

**Figure 1.** Category Tree of Inclusive Measures for Women with Disabilities: Institutional Resources, Academic and Work conditions, and Gender Perspective.

## 3. Results

The number of participants was determined by reaching data saturation. The total number of participants was 12 women belonging to the University of Seville community, as previously determined, had four students, five administrative staff, and three lecturers. Profiles and characteristics, or variables, are reflected in Table 1.

**Table 1.** Participant's profiles and characteristics.

| Interviewee | Profile | Age | Years at the US | Area of Knowledge | Type of Disability | Relatives with Disability | Belongs to Any Association |
|---|---|---|---|---|---|---|---|
| 1. María | Student | 46 | 2 | Health Sciences | Physical and mental | No | No |
| 2. Lola | Student | 49 | 2 | Social and Legal Sciences | Physical | No | Yes |
| 3. Carmen | Student | 26 | 5 | Social and Legal Sciences | Physical | No | Yes |
| 4. Esther | Student | 25 | 6 | Social & Legal Sciences | Hearing loss and deafness | No | No |
| 5. Juana | Lecturer | 39 | 21 (9 as PDI) | Social and Legal Sciences | Physical | Yes | No |
| 6. Ana | Lecturer | 46 | 23 | Engr. & Arch. | Others | No | Yes |
| 7. Marina | Administrative staff | 48 | 1.5 | Arts & Humanities | Mental | No | No |
| 8. Marta | Administrative staff | 47 | 15 | | Physical and mental | Yes | No |
| 9. Mercedes | Lecturer | 44 | 8 months | Engr. and Arch. | Physical | No | No |
| 10. Violeta | Administrative staff | 48 | 12 | Engr. and Arch. | Others | Yes | No |
| 11. Sara | Administrative staff | 62 | 35 | Social and Legal Sciences | Hearing loss and deafness | Yes | No |
| 12. Cecilia | Administrative staff | 47 | 14 months | Social and Legal Sciences | Physical | No | No |

*3.1. Institutional Resources: Resources Offered by the University of Seville*

As shown in Table 2, interviewees indicated that the resources offered by the University of Seville regarding their disability are insufficient, even more so when the individual needs any type of complementary resource to adapt their needs to university daily life. The situation derived from the coronavirus pandemic was an enabling factor for inclusion, as some of the interviewees observed how the virtual environment facilitated by the institution for their class follow-up or work performance met their needs much better. Thus, what for other students and workers was a limitation, for them was an advantage.

**Table 2.** Category "institutional resources" and its participants´ verbatims.

| Subcategories | Verbatims |
| --- | --- |
| Resources offered by the University of Seville | "Virtuality has somehow helped me to feel more 'normal', because all people were experiencing a similar situation, and now, I feel even less normal than before." (JUANA)<br>"Once you try online courses, comfortably at home, with mi headphones which makes everything quiet and nice. It is the best way to take a course." (SARA)<br>"I had no idea, and I do not think. . . It is not something they say to you once you enroll the university, like 'Hey, we have psychologists and jurists if you need them'. It is not something we remember to have been talked about." (CARMEN) |
| Communication and dissemination of available policies | "I think you must find out by yourself, as the University does not tell you that there is this department which helps with this or that. Basically, I entered the page [refers to Google search] and searching 'disability' at the University of Seville site, the results showed SACU [the university community student support service]." (LOLA)<br>"I had no idea, and I do not think . . . It is not something they say to you once you enroll the university, like 'Hey, we have psychologists and jurists if you need them'. It is not something we remember to have been talked about." (CARMEN)<br>"The general feeling is that everything is posturing, that is all. The feeling is like, 'We are principled, we are very aware'. But everything remains on the paper." (JUANA)<br>"You must be initiative-taking. There is no passive dissemination coming to you." (ANA)<br>"I think it would be necessary to inform about the different disabilities. For example, if in my unit I have a mental disability, we can have a seminar on metal disability attended by people with and without this type of disability and share our experiences . . .". (MARINA) |
| Difficulty in accessing resources. Complex bureaucratization of processes. | "She asked for it several times [a specific type of table] and finally she stopped signing up as the battle seemed a lost cause." (MARÍA)<br>"I have seen how this classmate in wheelchair is now in her third year and has been asking for a height-adjustable table since her first year, and three years after she is still waiting." (MARÍA)<br>"Besides, for example, a chair. I cannot use normal chairs because my body does not let me. It needs back support and a particular shape . . . When I informed the University, my Master tutor lent me her own office chair until I got one from social services. The chair from social services arrived just a month before I left, though." (CARMEN)<br>"A year ago, I asked if they could adjust one of the bathrooms of the professors' floor so I could avoid going down to the first floor every time, and they said me they would see how to do it. And it is still there, unadapted." (JUANA)<br>"The truth is that everything depends on people's goodwill" (. . .) "as no one knew who had to give the order, no one took any decision." (JUANA) |

### 3.2. Institutional Resources: Communication and Dissemination of Available Policies

Support policies available at the University of Seville are not conveniently disseminated among the community members, as participants commented in Table 2. In fact, interviewees perceive it necessary to conduct a complex search and consultation task in order to become familiar with the resources and initiatives that may be helpful in their daily lives.

As a complementary measure, training sessions and seminars on types of disabilities conducted by people belonging to the institution who could share their experiences with the rest of the university community are also suggested.

### 3.3. Institutional Resources: Difficulty in Accessing Resources. Complex Bureaucratization of Processes

Bureaucracy is a difficult and excruciating barrier when accessing resources, as timeframes are long and complex. Moreover, there is an added complexity of bureaucracy associated with the use of measures and resources available to students. Obtaining resources is also a limitation identified by both administrative staff and lecturers. The processes and procedures established to authorize measures aimed at women with disabilities inclusion are not clear enough.

### 3.4. Academic and Work Conditions: Communication of Special Needs

Women perceive, as Table 3 shows, that they must be continuously explaining their situation of disability for every procedure and every university year. There is no efficient communication within the University of Seville allowing all people involved in this process to be informed of the situation and requirements of these women, and therefore their common demand is that the procedure be streamlined. Even so, there are some instances of mistrust regarding capacities and abilities due to the situation of disability.

**Table 3.** Category "academic and work conditions" and its participants´ verbatims.

| Subcategories | Verbatims |
|---|---|
| Communication about special needs | "So, I would ask that every time someone with disabilities enrolls, it should be the university the one telling the department responsible, 'Hey, this is what needs to be done." (JUANA)<br><br>"Everything is fine, really. As I said before, I cannot complain about my colleagues or professors." (ESTHER)<br><br>"The truth is that the relationship and contact with mi colleagues has been excellent, I have had any issues with no one. However, when trusting me with different tasks maybe they have perceived me more insecure, and I have been relegated from some type of works, so this is something I have been observing, that they do not trust so much in my capacities." (MARINA)<br><br>"I feel constantly compensating. It is a mixture of pleasing and compensating people which is hard to manage. It is hard to comply with that all the time." (JUANA)<br><br>"And they may think my exam was easier when it is exactly the same and the grade scale as well. As if someone gave me the job. But I think they have already realized that I do the same work as everyone else, with no difference at all." (CECILIA) |

**Table 3.** *Cont.*

| Subcategories | Verbatims |
|---|---|
| Visible disability versus invisible disability | "Because if you force someone with disabilities to explain their situation again and again, you are not letting them to it leave aside and forget about it a single moment." (ANA)<br><br>"'So, what is what you have?', they say. And when I tell them my condition they say, 'Well, you seem to be fine ...'". [...] "It seems like you have to match this stereotype of sad, dying person. If they see you as positive, smiling... I have had a lot of issues regarding my appearance. Because I reflect my appetite life, as from the moment I wake up to the time I go to bed I am in pain, so I have to be really passionate about living, feeling, and making others feel fine. So, yes, I feel restricted just for not giving that impression." (ESTHER)<br><br>"Mine is not visible, so many people just do not get it, and only if you tell them, 'hey, I am disabled'" [...] "The problem is mainly that, when people realize you have a disability, because mine are in no way visible, not the physical or mental, so then they start asking, because they love asking, and they say, 'Oh, so you are disabled? Does not seem so' and 'So, what is it?' And it is really annoying because people think you are pretending, and in fact it is the other way round, never in my life I said, 'I cannot do this', no way, just the other way round." (MARTA)<br><br>"Sometimes, I also felt, now you have mentioned it, that when I just get off the car at the parking [exclusive for people con reduced mobility] people like, 'You do not seem to need the disabled parking space'. And this is why I even say to myself, 'I am going to force my limp to show them what is my condition' but then I think that if it is not hurting so much at that moment as to limp, why do I have to make obvious my disability, so people are satisfied? Do I have to justify myself in front of everyone?" (CECILIA) |
| Support among peers | "Very good, very good. As I've already told you, I can't complain about my colleagues or my teachers." (ESTHER)<br><br>"The truth is that the treatment from my colleagues has been excellent; I haven't had any problems with anyone. However, when it comes to trusting me for different tasks, they probably see me as more insecure, and of course, they ended up sidelining me from small jobs. That's what I really see, that they don't have much confidence in my abilities." (MARINA)<br><br>"And they think that my exam is easier when it's the same and when the grading scale is the same. And of course, it's as if they're just giving me the position. But they have realized that I work just like everyone else and I don't have any kind of difference." (CECILIA) |
| Academic and work conditions: institutional response | "Individualize support a bit more and focus on the needs of each of us" [...] "It should be generally acknowledged that there are as many disabilities as people with disabilities, and that no condition is identical to any other, so the aim should be to explore a bit more what people with disabilities really need at the University. That means that more individualized adaptations are needed according to the circumstances of each individual, and not by following a pattern that will never adjust to what each of us needs." (CARMEN)<br><br>"Each of us is different, and disabilities cannot be managed with a series of general policies, there must be a specialized team". (ANA) |

### 3.5. Academic and Work Conditions: Visible Disability versus Invisible Disability

The feeling that visible and objective disabilities are more believed to be dependable than other types of disabilities is shared by many of the interviewees, as shown in Table 3. Even so, there are some instances of mistrust regarding capacities and abilities due to the situation of disability. However, there seems to be a general sense that women with disabilities need to prove their competences much more than the rest, because sometimes their success is attributed to the measures they benefit from due to their disability. Consequently, they feel the urge to prove their own merits.

*3.6. Academic and Work Conditions: Support among Peers*

Support among peers is evident and declared among participants, as is shown in Table 3. They consider it an extra value and aid during their university years. No feelings of rejection are perceived on the part of their colleagues due to the resources received, but instead they perceive respect.

*3.7. Academic and Work Conditions: Institutional Response*

Interviewees declared, as Table 3 collected, that the University of Seville offers standardized procedures for all people with disabilities. However, this generates a complex situation, as every individual has different needs. This is why they demand greater individualization to meet everyone's requirements.

*3.8. Disability and Gender: Interrelation Being Female/Having Disabilities*

There is no general perception on the part of participants of the fact that being female places them in a dismissed or displaced position, and even less on the part of the institution, as is shown in Table 4. However, they are aware that this is very different within society, expressing that the discrimination they may suffer is precisely social, framed by the patriarchal society we live in.

**Table 4.** Category "Disability and gender" and its participants' verbatims.

| Subcategories | Verbatims |
|---|---|
| Interrelation being female or having disabilities | "I think both [women and men] are discriminated in the same way, not because women have more difficulties, no." (LOLA) <br> "Even more, I think that, maybe if I was a boy instead of a girl, with my academic background, my situation would be very different right now." (CARMEN) <br> "To be honest, I have not felt discriminated at the University on the basis of being a woman. Outside the University is different, I have worked in some places where you did not have the same chances." (MARINA) <br> "I think that the difference is more related to the type of disability than to gender, though maybe it is true that overprotection placed on women may affect you." (ESTHER) <br> "No, she is not managing the project because she will be constantly on leave" or "She is not going to coordinate this course because reasons" and I felt terrible. I have been saying no to things out of panic, I mean that my problem truly has been physical obligation, the rest does not matter to me. I do not mind working at night, I may be whacked and then working at 3 am. I am like that, but I need that flexibility. I have felt excluded from certain things, due to my disability." (ANA) <br> "Girls with disabilities have their freedom much more curtailed. I think it affects you completely, in your studies, relationships, aspirations . . ." [. . .] Besides, we must be aware of the fact that we are talking of people with disabilities in the context of an heteropatriarchy, a society with stereotypes shared also by people with disabilities, and particularly applied to women." (ESTHER) <br> "Normally, at the University, something I generally feel is that being a woman is like being very obliging, very collaborative, cooperative, and tending to show less of our character than me, who are less assertive. So, I wondered, with regard to my disability, what is my perception? And, apart from being obliging, which I believe it is associated to gender perspective, there is a part of the disability which tries to compensate. This means that, as you cannot certain things, you feel you must square in the sense that, 'I have been attending less classes than you, so I will finish that article' or 'I will submit it' or 'I will be second author'. I am always compensating. So, it is a combination, oblige and compensate, which is hard to balance." (JUANA) |

**Table 4.** *Cont.*

| Subcategories | Verbatims |
| --- | --- |
| Gender roles, reconciliation, and co-responsibility | "It is like the fact of being a woman will limit you to the care of your relatives or children because it is expected that you stay at home." (CECILIA)<br>"It is like women, particularly in the case of the University, are expected to be brilliant researchers, make a great professional career, struggle to reconcile, and if on top of that you add a disability, that in most of the cases it has to be very obvious to be recognized, as normally all support is scarce, even more if your disability is invisible. And you have always the feeling of having the imposter syndrome, which I personally never felt being haughty myself, but it is quite common to suffer from the imposter syndrome because you feel that everything you attain is been given to you." (ANA) |

Some participants expressed that disability affects them more than the fact of being a woman. Other participants consider that both limitations, being a woman and having a disability, add up.

*3.9. Disability and Gender: Gender Roles, Reconciliation, and Co-Responsibility*

A subcategory where care is part of the gender roles powerfully emerges, i.e., being a woman is associated with the role of caregiver above the condition of disability. This is expressed in Table 4 as a limitation in the performance of work or academic duties.

**4. Discussion**

The strategy for the rights of people with disabilities 2021–2030 is aimed at reducing the barriers that prevent them from participating in a full and effective way in all aspects of society and the economy [6]. The objective is that people with disabilities achieve individual autonomy, equal opportunities, involvement in society, freedom to make their own choices, and not suffer discrimination [1,7,8]. In this context, this study aimed to explore the situations experienced by women members of the University of Seville community during their years at the institution.

Higher university education may be a main form of preventive action against the social exclusion of people with disabilities, with positive results proven in the physical, cognitive, emotional, and social spheres [9]. As some studies have shown, students who complete their university studies see this situation reflected in an improvement in their quality of life [10].

Although in the University of Seville the integral program for people with disabilities [II Plan Integral de atención a las necesidades de apoyo para personas con discapacidad o con necesidad de apoyo por situación de salud sobrevenida] [11] is in force and has the objective of coordinating all the policies and practical guidance regarding disabilities and other health conditions so these people have equal opportunities, enjoy a secure, accessible and healthy work and study place, facilitating their integration in the labor market and society, it is however a fact that greater dissemination of this program is necessary to inform and sensitize, not just the people with any type of disability, but the university community as a whole.

Although most universities implement a positive action admission policy for students with disabilities, there is often no subsequent follow-up to ensure real inclusion, as effective resources to achieve equality within the university are lacking [12–15]. This has been identified throughout the research work, where participants pointed out precisely how necessary complementary resources were to facilitate their inclusion in the university, that bureaucracy was slowing down the processes, and, as a result, the required measures or resources were not easily achieved. In this sense, several studies highlight that universities make noted efforts to solve architectural problems [16], though other requirements may still need to be addressed.

Numerous sources agree with the experiences of the interviewees regarding the benefits of remote work towards fostering equality, as this practice relies completely on talent. Specifically, it confirms that people with disabilities benefited from this flexibility; people with reduced mobility needed fewer transfers; and blind and deaf people could adapt more easily. Clearly, remote work may help to reduce stress levels and improve well-being. However, it is important to maintain the balance and build social bonds to avoid isolation [17].

The demand of interviewees in favor of creating more sensitization strategies for the staff and personnel working with people with disabilities is aligned with the petitions of other entities, such as Fundación ONCE (2016), which consider it necessary to include within the educational programs offered by the University of Seville, particularly those related to teacher training [12,13,18], contents specialized in diversity and disability. This will allow those future professionals to be more aware of social diversity, which in turn will foster much more positive attitudes and behaviors towards the inclusion of people with disabilities [15,19–21].

Interviewees also pointed out the need to foster education in the academic community, as a lack of information on disability may contribute to strengthening the barriers to student inclusion [8]. To this problem area, it could be added the lack of knowledge about possible methodological strategies facilitating curriculum adaptation [6,13,18,22,23]. Moreover, there is not always complete awareness and sensitization on the part of the body of educators on the importance of university inclusive education [18], even though the most significant changes should be fostered by the professors involved in order to guarantee equal education. This lack of information or awareness is aggravated if there is no clear regulation raising awareness in society about the needs of this collective and the lack of institutional support [8]. Moreover, Martins et al. (2018) highlight that while staff attitudes towards the inclusion of students with disabilities are generally positive, there remains a prevalent perception of disabilities as deficits, underscoring the need for comprehensive changes to effectively adopt a social and educational model of disability [24].

This perspective aligns with the findings of Valle-Flórez et al. (2021), who analyzed the perceptions of faculty members regarding the inclusion of university students with disabilities. Their study highlights significant barriers related to accessibility, willingness to accommodate, and interaction dynamics, emphasizing the need for targeted training and policy development to address these issues. Effective inclusion must address multiple dimensions of identity and experiences of inequality to be truly impactful. Understanding women as a diverse group per se and focusing on functional diversity within higher education is essential to developing inclusive policies that recognize and address the complexities of inequality. Thus, it underscores the importance of enhancing visibility and developing agile, individualized policies to foster a truly inclusive educational environment. Notwithstanding, it is crucial to keep in mind that adaptations must be accomplished on a case-by-case basis. No general measures can be applied to cover all the individual requirements, as the particularity of each disability process makes it necessary to conduct an in-depth analysis of each individual's circumstances [12,25].

Some people declare that they find it more difficult to face discrimination experienced during their educational stages than their own disability, thus affecting their self-esteem and confidence and resulting in an increase in stress [14]. This can translate into students hiding their disability condition in order to avoid a situation of prejudice [26]. The fact that every year and on a daily basis, women's disabilities have to be made visible and explained sometimes leads to their concealment. Besides, people with non-visible disabilities feel more pressure to justify their circumstances, as participants largely declared.

From the perspective of the essential intersectional analysis, it is common to consider that being female is another constraint to be discriminated against in a situation of disability [3]. However, there is no agreement in this study regarding the following aspect: some women argue that discrimination suffered by women with disabilities is inherent in patriarchal society and the process of socialization, which puts more discriminatory

weight on the fact of being female; other participants expressed that disability is a more significant constraint; and the rest considered that it is the sum of the two. Similarly, a lack of consensus or disagreement has been found among the references; double discrimination has been historically asserted [2,3]. Nevertheless, other studies, such as the one conducted in New Zealand, surprisingly found that women with disabilities were not as disadvantaged in terms of employment, socioeconomic status, and domestic circumstances as men with disabilities. A particularly surprising finding was the fact that, in some instances, women with disabilities obtain better basic educational qualifications than both men with disabilities and women without disabilities [27]. This points to the need for thorough research on this double discrimination.

Being women, and despite their disability situation, participants are no exception to the gender imposition associated with care [28]. Disability aggravates the difficulties of reconciling personal, family, work, and academic life spheres. Although the University of Seville does not avoid its obligations as a co-responsible institution [29,30], in line with the social concern on matters of reconciliation and co-responsibility, it is probably necessary to give further consideration and importance to individuals, and particularly women with any situation of disability where difficulties to reconcile increase.

As originally planned, in order to increase validity, a triangulation was conducted on the information and bibliographical sources, interviews, and researchers from different fields, i.e., nursery, physiotherapy, social and cultural anthropology, psychology, and gender studies. The authors were also in contact with and debated with the team responsible for the care of people with disabilities at the University of Seville, a strength in the study that will help launch new adjustments in the strategies aimed at caring for people with disabilities in light of these results.

*Limitations*

Among the limitations found in this study, it is important to highlight the impossibility of approaching in depth all kinds of disabilities. There are probably other people with specific situations and concerns who require or would have expressed other types of needs. Likewise, given the fact that the sample only included women, it is not possible to establish a comparison between the experiences of men and women, which could have enriched this study. Another limitation that could be identified is that, due to COVID-19 restrictions, some interviews were conducted online while others were face-to-face. This could limit the expression of the interviewees.

## 5. Conclusions

The study underscores that although direct discrimination based on gender may not be overtly perceived, it operates within a patriarchal framework where such discrimination is embedded. To bridge the gender gap and foster the inclusion of women with disabilities, there is a need for more targeted efforts and policies. Peer support and respect are crucial protective factors for individuals with disabilities, especially women. Challenges that undermine an equitable and wholesome experience for people with special needs include a shortage of resources or barriers to access, a lack of understanding about disability issues, insufficient awareness within the university community, and a one-size-fits-all approach to disability support. Consequently, there is an imperative to enhance the visibility of resources. Therefore, further studies are essential to provide evidence that can lead to more effective interventions and policies.

**Author Contributions:** Conceptualization, F.L.-L., R.C.-M., C.A.-C. and L.S.M.; methodology, R.C.-M.; validation, F.L.-L., R.C.-M., C.A.-C. and L.S.M.; formal analysis, F.L.-L. and R.C.-M.; data curation, F.L.-L. and R.C.-M. writing—original draft preparation, F.L.-L. and R.C.-M.; review and editing, F.L.-L., R.C.-M., I.C.-G. and M.-L.B.-L.; supervision, R.C.-M.; funding acquisition, F.L.-L. All authors have read and agreed to the published version of the manuscript.

**Funding:** This research was funded by the Andalusian Government Equality Department, Consejería de Igualdad, Políticas Sociales y Conciliación. Instituto Andaluz de la Mujer. Junta de Andalucía.

**Institutional Review Board Statement:** The study was conducted in accordance with the Declaration of Helsinki and approved by the Ethics Committee of the University of Seville (protocol code 105-N-22).

**Informed Consent Statement:** Written informed consents were obtained from all participants involved in this study, in order to publish this paper. This research has passed approval by the ethics committee at the University of Seville (1015-N-22).

**Data Availability Statement:** The data supporting this study's findings are available from the corresponding author upon reasonable request.

**Acknowledgments:** We would also like to express our gratitude to all the participants of the study.

**Conflicts of Interest:** The authors declare no conflicts of interest.

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
