# Peer review of "Bridging Gaps, Fostering Inclusion: A Gendered Look at Disability Support for Women in Higher Education"

_women, doi:10.3390/women4030018_

Round 1
Reviewer 1 Report
Comments and Suggestions for Authors
Dear authors,
First of all, thank you for submitting the manuscript entitled “Bridging Gaps, Fostering Inclusion: A Gendered Look at Disability Support for women in Higher Education” for consideration for publication in Women. The manuscript fits the journal scope since it publishes research on women’s health, social determinants of health, and the healthcare system that serves women.
This intersectional research combines gender and diversity, explore the situation of women with disabilities in a specific context aiming to ascertain the effectiveness of inclusion measures at a public university in Seville.
Overall, the English language is appropriate and understandable.
The article is interesting, but there are a few points that need to be improved.
Although the topic has not yet been the subject of many studies, the authors could improve the introduction by including more articles on intersectionality and demonstrating the relevance of the topic and the pertinence of publishing studies like the one they are developing.
p. 58 – “2. Materials and Methods - 2.2. Participants” – in my opinion there should be a clear sentence with the participants’ descriptions. Number, age, role…
p. 97 – “Due to the COVID-19 pandemic, interviews 97 were conducted face-to-face by two collaborators” – what do you mean by that?
p. 121 - “Figure 1. Category tree” is repeated after and before the figure. In my opinion there should be a paragraph explaining the figure. On l. 154 there is some information regarding the figure but probably not in the right place.
Table 1 – why use the term: “Female student”? the readers already know that all the participants are female.
“PDI”; “PAS” what does that mean?
l. 270 – “Although most universities there is an admission policy” something is missing on this sentence.
Suggestions for future studies or practical suggestions could also be made in the conclusions.
Author Response
Dear authors,
First of all, thank you for submitting the manuscript entitled “Bridging Gaps, Fostering Inclusion: A Gendered Look at Disability Support for women in Higher Education” for consideration for publication in Women. The manuscript fits the journal scope since it publishes research on women’s health, social determinants of health, and the healthcare system that serves women.
This intersectional research combines gender and diversity, explore the situation of women with disabilities in a specific context aiming to ascertain the effectiveness of inclusion measures at a public university in Seville.
Overall, the English language is appropriate and understandable.
Response: Dear reviewer, thanks for your kind words.
The article is interesting, but there are a few points that need to be improved.
Although the topic has not yet been the subject of many studies, the authors could improve the introduction by including more articles on intersectionality and demonstrating the relevance of the topic and the pertinence of publishing studies like the one they are developing.
Response: We have updated the review, and have introduced new ideas that you can see in lines 28-39.
- 58 – “2. Materials and Methods - 2.2. Participants” – in my opinion there should be a clear sentence with the participants’ descriptions. Number, age, role…
Response: Thank you for your suggestion. Please, see lines 78-79.
- 97 – “Due to the COVID-19 pandemic, interviews 97 were conducted face-to-face by two collaborators” – what do you mean by that?
Response: Thank you, we changed the sense of the sentence with a new line introduced. We hope to make you seen now. Please see line 107.
- 121 - “Figure 1. Category tree” is repeated after and before the figure. In my opinion there should be a paragraph explaining the figure. On l. 154 there is some information regarding the figure but probably not in the right place.
Response: Thank you for this observation. Now is correct. The paragraph was moved the right location. Please, see lines 128-132.
Table 1 – why use the term: “Female student”? the readers already know that all the participants are female.
“PDI”; “PAS” what does that mean?
Response: Thank you for all recommendations. Female student was changed by student. PDI are lectures, and PAS are administrative staff. All terms were modified.
- 270 – “Although most universities there is an admission policy” something is missing on this sentence.
Response: The sentence has been revised for better clarity and flow. Please, see lines 276-278
Suggestions for future studies or practical suggestions could also be made in the conclusions.
Response: A new sentence following your suggestion was included. Please see lines 383-384.
Reviewer 2 Report
Comments and Suggestions for Authors
You may provide some practical guidelines (5. Conclusions), based on the results of your research project.
Comments on the Quality of English LanguageThe quality of the English language is very good. However, the text must be proofread.
Author Response
You may provide some practical guidelines (5. Conclusions), based on the results of your research project.
Response: Thank you for your observation. A new sentenced was included in this section. Please see lines 383-384.
Reviewer 3 Report
Comments and Suggestions for Authors
In this manuscript, the authors have investigated the effectiveness of inclusion measures for women with disabilities at a public university in Southern Spain. They found that no direct discrimination based on sex, but found resource scarcity, access challenges and lack of awareness as obstacles. They show that there is a need for enhanced visibility of inclusive measures and the development of agile, individualized policies. This study stresses raising awareness for non-visible disabilities through educational initiatives.
This manuscript provides interesting insights. The quality of the results presented is very good and supports the authors' conclusions. The experimental research component is technically and ethically sound. The title and the abstract of the manuscript are aligned with the conclusions. The figures and tables are adequate, and the language is clear and accessible.
Minor revision:
1. There is an arrow (from “Academic and work conditions”) showing no details in figure 1.
2. There is an extra “Y” is after physical in the type of disability in interviewee 1 and 8.
Author Response
In this manuscript, the authors have investigated the effectiveness of inclusion measures for women with disabilities at a public university in Southern Spain. They found that no direct discrimination based on sex, but found resource scarcity, access challenges and lack of awareness as obstacles. They show that there is a need for enhanced visibility of inclusive measures and the development of agile, individualized policies. This study stresses raising awareness for non-visible disabilities through educational initiatives.
This manuscript provides interesting insights. The quality of the results presented is very good and supports the authors' conclusions. The experimental research component is technically and ethically sound. The title and the abstract of the manuscript are aligned with the conclusions. The figures and tables are adequate, and the language is clear and accessible.
Response: Thank you for your positive feedback on our paper.
Minor revision:
- There is an arrow (from “Academic and work conditions”) showing no details in figure 1
Response: Thank you for your observation so helpful. Now is included the right information. Please, see line 134.
- There is an extra “Y” is after physical in the type of disability in interviewee 1 and 8.
Response: Thank you for your observation. Now is modified.
Reviewer 4 Report
Comments and Suggestions for Authors
This study investigates the effectiveness of inclusion measures for women with disabilities at a public university, aiming to understand their needs and experiences organized into categories assessing institutional resources, working/academic conditions, and the interplay of disability and gender.
In the Introduction section more attention should be given to the gap the study is trying to fill.
In the Materials and Method section the protocol of study is described in detail so that it can be reproduced.
In the Results section data were analyzed according to the tree categories: institutional resources, working/academic conditions, and the interplay of disability and gender.
In the Discussion section more engagement with the literature is needed.
The limitation subsection is incomplete, especially considering that some interviews were conducted face to face while others were online.
The Conclusion section does not offer openness for further studies.
Author Response
This study investigates the effectiveness of inclusion measures for women with disabilities at a public university, aiming to understand their needs and experiences organized into categories assessing institutional resources, working/academic conditions, and the interplay of disability and gender.
In the Introduction section more attention should be given to the gap the study is trying to fill.
Response: New sentences were introduced. Please see lines 28-39.
In the Materials and Method section the protocol of study is described in detail so that it can be reproduced.
Response: Thank you for the positive comments.
In the Results section data were analyzed according to the tree categories: institutional resources, working/academic conditions, and the interplay of disability and gender.
Round 2
Reviewer 1 Report
Comments and Suggestions for Authors
Dear authors,
I appreciate the significant effort you have invested in improving the overall quality of the manuscript and addressing the reviewers' questions and concerns. In my opinion the paper is now suitable for publication.
Author Response
Dear reviewer,
Thank you for your kind report.
Best wishes,
Reviewer 4 Report
Comments and Suggestions for Authors
The authors addressed the concerns of reviewers and improved the article, except for the introduction which still lacks sufficient information to be convincing,
Author Response
A new revision of the introduction was carried out.